# Social Connectedness in Schizotypy: The Role of Cognitive and Affective Empathy

**DOI:** 10.3390/bs12080253

**Published:** 2022-07-26

**Authors:** Jessica Stinson, Rebecca Wolfe, Will Spaulding

**Affiliations:** 1Department of Psychology, College of Arts and Sciences, University of Nebraska–Lincoln, Lincoln, NE 68588, USA; rebecca.wolfe@huskers.unl.edu (R.W.); wspaulding1@unl.edu (W.S.); 2South Texas Veterans Healthcare System, San Antonio, TX 78229, USA

**Keywords:** schizotypy, social connectedness, empathy, alexithymia, distress tolerance

## Abstract

Social connectedness is increasingly understood to be a resilience factor that moderates vulnerability to poor physical and mental health. This study examines cognitive and affective processes that support normal socialization and social connectedness, and the impact of schizotypy, in well-functioning college students. In this study, a total of 824 college students completed a series of self-report questionnaires, and structural equation modeling was then employed to identify relationships between cognitive and affective empathy, alexithymia, distress tolerance, social connectedness, and schizotypy. Schizotypy is a trait-like condition, presumed to be genetic in origin, associated with the risk for schizophrenia. Like schizophrenia, schizotypy is thought to have three distinct dimensions or categories, termed positive, negative, and disorganized. Results indicate that the respective dimensions of schizotypy have different pathways to social connectedness, through both direct and indirect effects. Positive schizotypy exerts a counterintuitive positive influence on social connectedness, mediated by positive effects on cognitive empathy, but this is obscured by the high correlations between the schizotypal dimensions and the strong negative influences on empathy and social connectedness of the negative and disorganized dimensions, unless all those intercorrelations are taken into account. Overall, the pathways identified by structural equation modeling strongly support the role of empathy in mediating the impact of schizotypy on social connectedness. Implications for the etiology of social impairments in schizotypy, and for interventions to enhance social connectedness to improve quality of life and reduce health disparities in people at risk for severe mental illness, are discussed.

## 1. Introduction

Recent interest in health disparities associated with severe mental illness (SMI) has included a focus on health-related behavior, with the premise that poor health is, at least in part, a consequence of the effects of SMI on behaviors that normally constitute protective factors. Much attention has been given to behaviors that are most proximal to health consequences, such as those that support a healthy diet and lifestyle, and those whose failure creates health risks, such as cigarette smoking and substance abuse. Comparably proximal, and potentially equally important, are social and interpersonal behaviors. Social support and affiliation tend to moderate the impact of health problems and vulnerabilities [1,2,3,4,5], and deficits in social connectedness compound that impact, including increased stigma and chronic states of loneliness that further decrement quality of life [4,6,7,8]. Social connectedness refers to people’s general sense of belonging and interpersonal closeness in their social world [9]. It is arguably presumed to reflect the collective influence of social-cognitive and emotional self-regulation processes on a person’s global social functioning, especially the condition of their social support system. Consideration of mechanisms and failures of social connectedness should be included in a comprehensive approach to addressing health disparities and health behaviors in SMI.

Studies of *subclinical populations*, i.e., those with key features common to frank mental illness, have been useful in identifying etiological processes important to the development of the frank disorder. For SMI, the construct of *schizotypy* serves to identify one such population. When severe enough to meet the criteria for schizotypal personality disorder, it is considered part of the schizophrenia spectrum. With substantial overlap between clinical and subclinical populations, research on schizotypy has become an important organizational framework for the study of schizophrenia spectrum disorders [10].

Individuals with elevated levels of schizotypy show patterns of abnormality in neurodevelopment similar to those associated with schizophrenia [11], as well as psychosocial risk factors, including urbanicity; poverty; discrimination associated with minority status; immigration; poor parental communication; poor parental care; and various types of childhood adversities such as abuse, neglect, and bullying [12]. Schizotypy is associated with impairment in social competence, rapport with family and friends, interpersonal engagement, social and recreational activity, and occupational and academic functioning [13,14]. Many of the social cognition and social functioning deficits observed in subclinical groups are thought to predate and predict the onset of the frank disorder [15,16].

Deficits along the schizotypy continuum present an opportunity to study schizophrenia spectrum disorders without many of the challenging confounds faced when studying clinical samples, including severe symptoms and psychosis, distress, comorbidity, and psychotropic medication [12]. Research on schizotypy also provides unique opportunities to study the onset and course of schizophrenia spectrum disorders. Early detection and intervention have become central concerns of research and practice [17], with the promise of inflecting the natural course of the disorder, producing better long-term outcomes [18,19]. Schizotypal symptoms and impairments play a key role in developing clinical practices for these purposes. The present study evaluates some key impairments in social cognition common to both subclinical schizotypy and schizophrenia spectrum disorders, toward an understanding of those impairments in the health disparities observed in SMI.

## 2. Background

### 2.1. Subtypes and Dimensions of Schizotypy

Schizotypy is a set of traits that are qualitatively similar to features of schizophrenia spectrum disorders and fall along a continuum of severity in a normative population [20]. Schizotypy is considered a vulnerability for schizophrenia spectrum disorders [21]. Like schizophrenia, schizotypy includes three dimensions or categories of symptoms and related features: *positive*, *negative*, and *disorganized*. The positive features of schizotypy are abnormalities in content of thought, perceptual abnormalities, and paranoia, whereas negative features include alogia, anergia, avolition, anhedonia, and flat affect [22]. The disorganized features refer to difficulties in organizing and expressing one’s thoughts and behaviors [22]. Each category can be measured on a continuous dimension of severity, and schizotypy can be understood as a profile of the three dimensions. Much of the research on schizotypy has not considered the importance of these dimensions. This study evaluates the differential relationships of those dimensions with impairments in personal and social functioning relevant to health disparities.

### 2.2. Psychopathology, Complex Cognition, and Social Connectedness

Deficits in complex social cognition and interpersonal relationships are logically a source of social isolation and disconnectedness, and are often implicated in psychopathology (e.g., [23,24]). *Empathy* is one facet of social cognition that has long been of interest in research and clinical practice. Psychotherapy and related interventions often target various aspects of empathy for the purpose of improving social functioning [25,26,27,28]. However, the precise nature of the relationship between empathy and various forms of psychopathology is not well understood. This is especially true of schizophrenia spectrum disorders and individuals at risk for later development of these and other disorders associated with SMI. Although there is considerable support for the importance of empathy in successful social functioning [23,29], much of the research on empathetic deficits in SMI has yielded inconsistent or unclear results. Research on conditions that incur *risk* for SMI, including schizotypy [14,29,30,31,32,33], remains quite sparse and methodologically limited. This is especially problematic because a greater understanding of vulnerable states such as schizotypy is an important conduit to understanding the processes of SMI itself.

### 2.3. Gaps in Current Understanding of Empathy and Social Functioning

The failure of previous research to consider schizophrenia and schizotypy as multidimensional constructs, with differential deficits across subtypes, accounts for some of the inconsistencies in research findings on empathy in these populations. A related problem lies in inconsistent operational definitions of empathy, usually conceptualized as *either* a complex cognitive process or an emotional response to social stimuli. Understanding *both* cognitive and affective aspects of empathy, and the relationships of these aspects to one another, is needed to more fully understand the impairments in socialization and social connectedness observed in individuals at risk for poor physical health associated with SMI.

Earlier psychological research conceptualized empathy as a strictly cognitive process that emphasized being able to take the perspective of another (e.g., [34]). More recently, conceptualizations of empathy tend to include both cognitive and affective dimensions [35,36,37]. Decety and Moriguchi [23] defined empathy as “the capacity to *share* and *understand* the emotional states of others in reference to oneself” [emphasis added]. By their definition, empathy involves not only the emotion match, but also the cognitive ability to understand the context of what the other person is feeling. Although this two-fold definition has become the modal theoretical approach to the study of empathy, few existing measures of empathy appropriately take this multidimensional perspective.

*Cognitive empathy* is the ability to appraise and represent the emotional states of others [37,38,39]. It is considered by many to be synonymous with *perspective taking* and *theory of mind* [35,38,39,40,41,42,43], or at least closely related [24]. Perspective taking is the ability to apprehend another’s point of view while maintaining a distinction between self and other [44,45]. Theory of mind (ToM) refers to the ability to reason and make attributions about the mental states of others, including their thoughts, motivations, and beliefs [46,47]. ToM has also been used to describe *emotional* perspective taking [48]. Cognitive empathy facilitates recognition of others’ emotions, almost by definition, but it is important to note that one can apprehend others’ perspective or make attributions about their mental states without experiencing the same emotion, or any emotion at all. Cognitive empathy provides the “understanding” component of empathy, but additional affective information is required to achieve the “sharing” component.

Individuals with schizophrenia are consistently shown to demonstrate impairments in cognitive empathy [45,49,50,51,52,53,54,55,56], with some early evidence suggesting this may be specific to negative [53,54,56] and disorganized subtypes [57]. Existing, but limited, research on cognitive empathy in schizotypy generally supports deficits among negative subtypes, though some evidence for deficits among those with positive schizotypy has also been reported [32,33].

The emotional component of empathy is often termed *affective empathy* and specifically focuses on an observer’s emotional response to another’s situation [23,24,29]. Affective empathy is thought to reflect a vicarious sharing of emotions, or a matching of the observer’s emotion to the other’s emotion [58]. Consistent with this idea, another term that is often used interchangeably with affective empathy is *emotion contagion*, also described as an individual’s unconscious tendency to imitate the emotions of others [59].

A significant criticism of existing multidimensional measures of empathy is that they often conflate affective empathy with sympathy or personal distress. Although previous studies often consider affective empathy to be synonymous with emotion contagion or emotion-matching [58,60,61,62,63], many of the most commonly used measures of empathy do not account for this mimicry of emotion in their conceptualization and measurement. In fact, these authors are aware of only one multidimensional scale of empathy, the Questionnaire of Cognitive and Affective Empathy (QCAE; [42]), that was constructed for adult populations and appropriately utilizes an emotion-matching conceptualization of affective empathy. This measure has been employed in two known studies with individuals with schizophrenia, providing early evidence of heightened emotion contagion among this population [54,57], though no known studies have examined true emotion contagion in schizotypy.

*Emotional responsiveness to others* [42] is a construct that serves to complete the integration of cognitive and affective information into a fully empathic response. Impairment in responsiveness to others is characteristic of schizophrenia spectrum disorders and is vulnerable to elevated personal distress and deficits in perspective-taking [24,64]. Obviously, the construct is also associated with social connectedness, and is therefore a logical candidate as a mediator of the impact of other components of empathy.

### 2.4. Developmental Factors: Alexithymia and Distress Tolerance

Two additional constructs deserve consideration because they represent processes that appear earlier in human development, support empathy processes that develop and operate in later childhood and adolescence, and are observed to be impaired in schizophrenia spectrum disorders. Therefore, they are logical candidates as mediators of the impact of schizotypy on social connectedness. *Distress tolerance* is a type of emotion regulation: the ability to maintain functioning despite elevated personal distress. Deficits in distress tolerance and related processes are observed in schizophrenia spectrum disorders [65,66,67]. In addition, distress tolerance is a trait-like ability that moderates the confounding effects of personal distress, giving it particular methodological relevance to assessing empathy. *Alexithymia* is a decreased or absent ability to label and describe one’s own emotional state [68], of interest in clinical research due to its significant relationship to psychopathology, and is associated with social dysfunction [69,70,71]. Elevated levels of alexithymia have been observed in schizophrenia populations [72,73] and individuals high in schizotypal traits [68,71,74]. Greater levels of alexithymia are associated with poorer empathetic ability [68,74,75,76].

### 2.5. Study Aims and Hypotheses

The purpose of this study is to articulate the relationships between the various dimensions of schizotypy, quantitative measures of cognitive and affective empathy, related dimensions of social cognitive and affective functioning, and social connectedness, using the methods of structural equation modeling. The goals are to clarify the role of empathy in social impairments associated with schizotypy, and to identify potential targets for treatment intended to improve socialization and social connectedness, thereby reducing their impact on health vulnerabilities, in individuals with SMI and at risk for the future onset of SMI. Taken together, previous findings on empathy and related constructs in schizotypy generate the following hypotheses:Higher positive, negative, and disorganized schizotypy are all associated with poorer social connectedness.Higher positive, negative, and disorganized schizotypy are all associated with poorer distress tolerance.
A.Poorer distress tolerance is associated with poorer perspective taking and less responsiveness to others, but with higher emotion contagion.
Higher positive, negative, and disorganized schizotypy are all associated with higher levels of alexithymia.A.Higher levels of alexithymia are associated with worse performance on perspective taking, less responsiveness to others, and less emotion contagion.
Higher negative and disorganized schizotypy are associated with poorer self-reported cognitive empathy (when cognitive empathy is comprised of both perspective taking and efforts to represent others’ mental states).
A.The relationship between high negative/disorganized schizotypy and lower cognitive empathy is mediated by decreased levels of distress tolerance and increased levels of alexithymia.
Higher negative and disorganized schizotypy are associated with greater emotion contagion.
A.The relationship between high negative/disorganized schizotypy and greater emotion contagion is mediated by distress tolerance, alexithymia, and cognitive empathy.
Higher negative and disorganized schizotypy is associated with deficits in general responsiveness to others’ feelings.
A.The relationship between high negative/disorganized schizotypy and lower emotional responsivity is mediated by distress tolerance, alexithymia, and cognitive empathy.
The relationship between negative/disorganized schizotypy and social connectedness is mediated by abnormalities in cognitive empathy, emotion contagion, and responsiveness to others’ feelings.

The specific hypotheses are combined in the complete model of causal and mediating relationships diagrammed in Figure 1. The assumptions of causal primacy, proceeding from left to right in the model diagram, are based on an a priori chronological and developmental sequence: (1) schizotypy is a genetically-influenced trait-like condition present at the earliest stage of individual development, but can be identified and measured via self-report in adults; (2) distress tolerance and alexithymia reflect emotion regulation processes that develop in early- to mid-childhood, and they are correlated but neither has causal primacy over the other; (3) emotion contagion (affective empathy), cognitive empathy, and responsiveness to others are processes of complex social cognition that develop in later childhood and adolescence, in the presence of existing distress tolerance and the processes whose impairments produce alexithymia; (4) cognitive empathy has causal primacy over emotion contagion and responsiveness to others, in accordance with the generally understood role of cognitive attributional processes in emotional experience [58], emotion contagion and responsiveness to others are correlated, but neither has causal primacy over the other; (5) social connectedness is a product of interaction between pre-existing schizotypal status, emotion regulation, and social cognition factors, and the person’s developing social world.

## 3. Methods and Materials

### 3.1. Participants

Participants were initially 844 undergraduate students recruited online from a recruiting pool of students enrolled in an introductory psychology course. For participating, students received course credit in the form of one credit per half hour of participation. No screening was performed for elevated schizotypy or mental health history. The distributions of these indicators in this sample are presumed to be the same as for university undergraduates in general.

Informed consent and procedures were followed in accordance with a protocol approved by the Institutional Review Board of the University of Nebraska-Lincoln. To protect data integrity, 3 items were embedded in the questionnaires to detect inattention or random responding. Of the initial 844 participants, 20 were excluded from further analysis due to failing to pass at least 2 out of 3 validity items. This left a final study sample of 824, whose responses were included in the multivariate analyses.

### 3.2. Procedures

Participants were invited to take part in an online survey study in which they were asked to complete self-report measures including social desirability responding, demographics, measures of schizotypy, cognitive and affective empathy, distress tolerance, alexithymia, social connectedness, and impacts of the COVID-19 pandemic on well-being and areas of daily functioning.

### 3.3. Measures

Except for the demographics, all the measures employed in this study are reflective psychometric measures reflecting an underlying psychological construct, therefore requiring confirmation of interitem reliability in the form of Cronbach’s alpha [77]. Although only instruments with established interitem reliability were selected, their interitem reliability in the present sample was confirmed before inclusion in subsequent analyses. Cronbach’s alpha was computed with SPSS Statistics version 26 [78].

*Demographics*. A basic demographics survey collected information about participants’ race, ethnicity, sex at birth, gender, age, academic status, socioeconomic background, and psychiatric history. Sex at birth was of particular interest as a potential control variable, as research consistently indicates higher levels of self-reported empathy among women [42,79,80,81].

*Schizotypy*. The Multidimensional Schizotypy Scale (MSS; [21]) was used to collect information about participants’ experience with subclinical psychotic-like experiences. The MSS is a newer 77-item self-report measure of schizotypy that provides updated question wording, conceptualizations of schizotypy, and improved psychometric properties [21]. Its factor structure allows for the reliable measurement of positive, negative, and disorganized features of schizotypy, thus better capturing the heterogeneity of schizotypal presentations. Higher scores in each domain reflect higher levels of schizotypy. Possible scores range from 0 to 26 for positive schizotypy, 0 to 26 for negative schizotypy, and 0 to 25 for disorganized schizotypy. In this study, Cronbach’s alpha was 0.885 for positive schizotypy, 0.858 for negative schizotypy, and 0.935 for disorganized schizotypy.

*Empathy*. The Questionnaire of Cognitive and Affective Empathy (QCAE; [42]) is a validated self-report measurement of affective empathy, cognitive empathy, and emotional responsiveness to others. It has demonstrated good convergent validity, construct validity, and internal consistency [42]. The cognitive empathy component of the measure is comprised of a perspective taking subscale (i.e., ability to take on another’s perspective) and an online simulation subscale (i.e., effort given to understand and mentalize another’s emotional state). The affective empathy component allows for the separate measurement of affective responses consistent with sympathy/general responsiveness to others, as well as the measurement of emotion-matching. This affective empathy component is comprised of the emotion contagion subscale (i.e., emotion matching to a target), a proximal responsivity subscale (i.e., emotional responsiveness to the moods of close others), and a peripheral responsivity subscale (i.e., emotional responsiveness to the moods of distant others). In this study, proximal and peripheral responsivity were combined to represent general emotional responsivity to others. Participants rated their agreement with items on a 4-point Likert-type scale, where higher total scores reflect greater self-reported empathy. Possible scores range from 19 to 76 for cognitive empathy and 4 to 16 for emotion contagion. Of note, in this study, item 17 of the responsiveness to others subscale was dropped in order to achieve an acceptable Cronbach’s alpha of 0.711. Therefore, scores for this variable had a possible range of 8–32. Cronbach’s alpha was 0.902 for cognitive empathy and 0.741 for emotion contagion.

*Distress Tolerance***.** Distress tolerance was measured using The Distress Tolerance Scale (DTS; [82]). The DTS is a 15-item self-report measure of one’s ability to tolerate emotional distress and has shown good test-retest reliability, internal consistency, and construct validity [82]. Participants rated their agreement with items on a 5-point Likert-type scale, where a higher total score reflects a greater tolerance of distress. Possible scores range from 15 to 75. In this study, Cronbach’s alpha was 0.920.

*Alexithymia*. Alexithymia was measured using the 20-Item Toronto Alexithymia Scale (TAS-20; [83]). This is a 20-item self-report measure with demonstrated good psychometric properties. It captures 3 dimensions of alexithymia: difficulty identifying one’s own feelings, difficulty describing one’s own feelings, and externally-oriented thinking [83]. Participants rated their agreement with items on a 5-point Likert-type scale, where a higher total score reflects greater degrees of alexithymia. Possible scores range from 20 to 100. In this study, Cronbach’s alpha was 0.859.

*Social Connectedness.* Social connectedness was measured by The Social Connectedness Scale-Revised (SCS-R; [9]). This is a 20-item self-report measure that assesses an individual’s general sense of belonging and interpersonal closeness with the social world. This measure was selected because many other widely used measures of social functioning include behavioral items that were not appropriate given the context of the ongoing COVID-19 pandemic. The SCS-R has demonstrated good convergent and divergent validity, as well as internal consistency [9]. Participants rated their agreement with items on a 6-point Likert-type scale, where a higher total score reflects greater degrees of social connectedness. Possible scores range from 20 to 120. In this study, Cronbach’s alpha was 0.935.

*Social Desirability Responding*. To control for social desirability responding, particularly on measures of empathy, the Marlowe–Crowne Social Desirability Scale-Short Form Version C [84] was used in this study. This measure consists of 13 true/false items aimed at identifying participants’ tendency to respond to self-report survey items in a socially desirable manner. The Short Form Version C of this scale has demonstrated good reliability and validity [84]. Higher scores on this measure represent greater social desirability responding. Possible scores range from 0 to 13. In this study, Cronbach’s alpha was 0.670. Internal consistency was not improved with the removal of any item. It is controversial whether alphas in the 0.6–0.7 range are acceptable. For the purposes of this study, the observed alpha was deemed sufficient to use the measure as a control variable in the path analyses.

*Social Impact of the Current COVID-19 Pandemic*. As this study was conducted during an ongoing health-related pandemic that has resulted in widespread quarantining and physical distancing, care was taken to control for the possible influence of these factors, particularly on social connectedness. This study asked participants to complete a measure of how the pandemic has impacted them across various domains. This measure, the Controlling for Environmental Confounds Questionnaire, is a set of 21-items that assesses the potential influence of COVID-19 across multiple domains related to mental health and social relationships. This set of questions was recently designed by members in our lab in order to help control for recent events that may have implications on study data. It has not been used in previous studies and its inclusion in this study is justified by its face validity. This study specifically examined the endorsement of COVID-19 related changes in various types of interpersonal relationships. This included items on family, friends, romantic relationships, and coworkers, as well as overall social life. Participants rated the extent to which the pandemic has impacted these areas of functioning, with higher scores reflecting greater disruption. These five items were aggregated into a single variable representing overall impacts of the pandemic on social connectedness to others. Possible scores ranged from 0 to 20. In this study, Cronbach’s alpha was 0.780. This measure was entered in the modeling analyses as a control variable.

### 3.4. Data Analysis

*Bivariate Associations*. Bivariate correlations were used to test associations between the respective study measures, including the 3 dimensions of schizotypy, distress tolerance, alexithymia, cognitive empathy, emotion contagion, responsiveness to others’ emotions, and social connectedness. Bivariate correlations were also used to examine associations between study variables and potential control variables (social desirability responding and effects of the pandemic on social connectedness), while independent samples’ *t*-tests were used to examine sex differences. All measures were confirmed to be normally distributed by an examination of skewness and kurtosis. The Bonferroni correction for *p* < 0.05 was utilized for all bivariate analyses. These data were analyzed using SPSS Statistics version 26 [78].

*Path Model*. For the complete path model of this study, structural equation modeling was employed. Data were analyzed using Mplus [85]. The hypothesized model is a *saturated* path model, with an equal number of known and estimated parameters, and no latent variables. The inclusions of assumptions about causal direction and hypotheses of indirect effects make it a *mediation* model, for which the SEM path analysis is a suitable analytic tool. Saturated path models are most suitable for the purpose of this study, identifying specific direct and indirect effects between variables, as opposed to optimizing the model’s predictive accuracy or evaluating latent variables [86].

To evaluate parameter estimates, mediation was tested using a maximum likelihood estimation with bootstrapping [87]. Bootstrapping provides an empirical approximation of sampling distributions of indirect effects, maximizing power, and minimizing Type I error. Bias-corrected bootstrapping with 10,000 resamples drawn to gain the 95% confidence intervals for the indirect effects was used. Bootstrapping can also address the non-normality of data; although, all data in the current study were normally distributed. A maximum likelihood estimation addresses any issues with missing data. In the present study, there was a very small amount of missing data (e.g., covariance coverage for the primary mediation model ranged from 0.999 to 1.000).

## 4. Results

*Demographic and Descriptive Measures*. The demographic characteristics of the sample and the parameters of the study measures are summarized in Table 1 and Table 2. The demographics were consistent with those expected in a large American midwestern university. Parameters of the study measures were within expectation based on previous research with these measures.

*Correlations among Study Measures and Control Variables*. Correlations between the measures of schizotypy, empathy, distress tolerance, alexithymia, and social connectedness are shown in Table 3. For the 36 bivariate correlations in the matrix, *p* < 0.0014 survived a Bonferroni correction for *p* < 0.05. There was substantial intercorrelation, with 31/36 reaching statistical significance. Only 6 of the 31 significant correlations were between the subscales of a single instrument. The directions of the significant correlations were in the expected direction, with one notable exception: higher negative schizotypy was associated with *lower* emotion contagion.

For the 18 bivariate correlations between the study measures and the continuous control measures (social desirability and pandemic effect), *p* < 0.0028 survived a Bonferroni correction for *p* < 0.05. For the social desirability control measure, three correlations reached significance, with cognitive empathy (r(820) = 0.151, *p* < 0.001), distress tolerance (r(820) = 0.309, *p* < 0.001), and social connectedness (r(820) = 0.311, *p* < 0.001). For the pandemic impact control measure, seven correlations reached significance, with positive schizotypy (r(820) = 0.194, *p* < 0.001), disorganized schizotypy (r(820) = 0.190, *p* < 0.001), emotion contagion (r(820) = 0.221, *p* < 0.001), responsiveness to others (r(820) = 0.169, *p* < 0.001), and alexithymia (r(818) = 0.158, *p* < 0.001). Social impact of the pandemic negatively correlated with distress tolerance (r(819) = −0.226, *p* < 0.001), and with social connectedness (r(819) = −0.199, *p* < 0.001).

For the nine *t*-tests between sex at birth and the study variables, *p*< 0.006 survived a Bonferonni correction for *p* < 0.05. Females reported significantly lower negative schizotypy (t(820) = 3.001, *p* < 0.003. Females had significantly higher scores on all facets of empathy, cognitive empathy (t(820) = 2.767, *p* < 0.006), emotion contagion (t(820) = 6.603, *p* < 0.001), and responsiveness to others (t(820) = 7.099, *p* < 0.001). Females’ lower distress tolerance did not survive the Bonferroni correction (t(819) = 2.355, *p* > 0.019).

In summary, the control variables have modest but statistically significant correlations or mean differences with 14 of the study measures, supporting their inclusion in the path model analyses.

*Complete Path Model:*Figure 2, Figure 3 and Figure 4 show the pathways between the schizotypy dimensions and social connectedness. The pathways from the respective schizotypy dimensions are shown separately, to facilitate visual inspection, but it is important to note that all the pathways take into account the intercorrelations of all the study measures and the control measures together. The one complete model includes all three schizotypal dimensions. There are several pathways among the other study variables common to more than one dimension.

The complete path model reveals an unexpected and counterintuitive pathway, a positive effect of positive schizotypy, mediated by cognitive empathy, on emotion contagion, responsiveness to others, and social connectedness. This pathway was not evident in the bivariate correlations, where the correlations between the schizotypy dimensions and the larger correlations between the schizotypy dimensions and the other measures masked this effect.

The pathways shown in Figure 2, Figure 3 and Figure 4 represent statistically significant direct effects between the respective measures. Table 4 shows the path coefficients and confidence intervals for the significant indirect (mediated) effects.

*Summary of Results*. The three dimensions of schizotypy, positive, negative, and disorganized, had different patterns of correlation with measures of distress tolerance, alexithymia, emotion contagion, cognitive empathy, responsiveness to others, and social connectedness. When the effects of sex at birth, social desirability, and the pandemic were controlled, and the correlations were considered in the context of an a priori causal sequence, as in the hypothesized path model, there were different pathways between the respective schizotypy dimensions and social connectedness.

A small but significant unexpected and counterintuitive positive effect of positive schizotypy on social connectedness appeared, mediated by a direct effect on cognitive empathy. There was no direct effect of positive schizotypy on social connectedness, but there was a direct effect on responsiveness to others; although, that had no effect in turn on social connectedness. Negative and disorganized schizotypy also followed different pathways to social connectedness. All the valences of the path coefficients were in the expected direction, with two exceptions: negative schizotypy exerted a *negative* effect on emotion contagion through an indirect effect on alexithymia and cognitive empathy; and disorganized schizotypy had a net *positive* effect on responsiveness to others as a result of its strong impact on distress tolerance, reflecting increased sensitivity to others in distress. Although not hypothesized, the two exceptions are not counterintuitive, and consistent with previous findings on the complexity of interrelationships between measures of distress, emotion regulation, and sensitivity to others.

Of the three measures of empathy, cognitive empathy exerted a direct effect on social connectedness, but emotion contagion and responsiveness to others did not. The two empathy-related emotion regulation measures, distress tolerance and alexithymia, had different mediating roles in the respective schizotypy dimensions.

On a case-by-case basis, the positive effects of positive schizotypy were usually obscured by the larger correlations between the respective schizotypy dimensions and the negative correlations between schizotypy and the other study measures, as they were in the bivariate correlations. Negative schizotypy did exert a direct negative effect on social connectedness, plus indirect effects mediated by cognitive empathy and alexithymia. Disorganized schizotypy also exerted a direct negative effect on social connectedness, plus additional effects mediated by alexithymia, cognitive empathy, and distress tolerance. All three schizotypy dimensions exerted direct or mediated effects on emotion contagion and responsiveness to others, but these did not impact social connectedness.

## 5. Discussion

The results of this study support the hypotheses as diagrammed in Figure 1, of causal pathways between schizotypy and social connectedness, mediated by processes associated with empathy. The pathways and mediators associated with the three dimensions of schizotypy are different. This is not inconsistent with the expectation based on previous research on schizotypy and empathy; although, the previous research does not provide the specificity that the current findings reveal. There is one counterintuitive finding, that when all the other intercorrelations between the study measures and the control measures are taken into account, there is a small but significant *positive* effect of positive schizotypy on social connectedness, mediated by cognitive empathy, and a positive effect on responsiveness to others, which is not passed on to social connectedness. The counterintuitive finding may resolve inconsistencies in previous research regarding the relationships of schizophrenia spectrum disorders and processes associated with empathy, when the subtypes of schizotypy or schizophrenia are not distinguished, and/or the distinctions between emotion matching, distress tolerance, and interpersonal responsiveness are not taken into account.

As expected, the results provide some important clues about the etiology of schizophrenia spectrum disorders, and about the role of social functioning in psychological morbidity and health disparities. They suggest that schizotypy, as a congenital condition, creates vulnerability for psychopathology by compromising the subsequent development of cognitive and affective processes associated with empathy, and in turn, social connectedness. Social isolation compounds the morbidity of schizophrenia spectrum disorders, and also compromises the resilience to health disparities provided by social connectedness.

The results also provide some important clues about the optimal targeting of cognitive impairments, for the purpose of reducing their effects on social functioning and health. Positive schizotypy without accompanying elevations of negative and disorganized schizotypy may be rare, but when it occurs, it may represent a paradoxical resiliency factor, perhaps stimulating compensatory effects on social cognition and behavior. Patients with high positive schizotypy and low negative and disorganized schizotypy may benefit optimally from a therapeutic approach emphasizing the application of relatively normal cognitive empathy and responsiveness to others, rather than the normalization of cognitive or affective impairments. For most patients, however, the pathways from negative and disorganized schizotypy to social connectedness demand therapeutic attention to impairments in distress tolerance, alexithymia, and cognitive empathy, that mediate the impact on social connectedness. Therapeutic modalities have been developed whose targets include some processes measured in this study, including distress tolerance [88,89,90], alexithymia [91,92], and cognitive empathy [93,94]. All could be usefully adapted for people at risk for schizophrenia spectrum disorders. There may also be a need for new modalities targeting emotion contagion (affective empathy) and responsiveness to others.

### Study Limitations

The primary limitation of this study is in the validity of the a priori developmental sequence of the study’s key constructs: schizotypy, processes associated with empathy, and social connectedness. The data are cross-sectional in the sense that they were all collected at one time point, and causal inferences cannot be drawn from cross-sectional data with absolute confidence. However, the constructs measured are ordered in a temporal, sequential, and developmental sequence. It is generally understood that schizotypy is a congenital condition that can be identified through self-reports in adults, and that adult self-reports can measure cognitive and affective processes like distress tolerance and emotional contagion, whose parameters were established in earlier periods of development. Nevertheless, the alternative view cannot be dispositively rejected, that even if schizotypy is something present at birth, a self-report in adulthood is the product, not the origin, of its effects on development. At this stage of the science, confidence in a causal interpretation of the present data is subject to confidence in the construct validity of schizotypy and the processes associated with empathy. Short of actual longitudinal data, the construct validity of the relevant variables is generally understood to be a legitimate consideration in causal assumptions and mediated relationships in structural equation modeling, especially in developmental and healthcare applications [95,96,97,98].

Ultimately, longitudinal studies of developing empathy-related processes in healthy and vulnerable individuals will be necessary to confirm the sequence in which those impairments and vulnerabilities appear. Considering the multitude of findings of impaired cognition and affect regulation in children at risk for schizophrenia spectrum disorders (e.g., [99]), such confirmation would hardly be surprising. On the other hand, such studies may not resolve the sequential relationships between early cognitive development and the construct of schizotypy. Since schizotypy is traditionally identified via adult self-report (either through questionnaire or clinical assessment), it is unclear whether it can be identified or measured in childhood. The degree to which children at risk for adult-onset schizophrenia, as determined by genealogy or by developmental impairments, are the ones later identified as schizotypal via adult self-report, is unknown. It may be that schizotypy as currently identified in adults must be deconstructed into separate developmental components, genetic, cognitive, affective, and socio-behavioral, to be studied in childhood and adolescence.

Although the sequential relationships of schizotypy and processes associated with empathy are somewhat conjectural, that is much less the case with social connectedness. It is far less plausible to hypothesize that social disconnectedness in adults is the cause rather than the effect of the trait-like features of schizotypy and impairments in cognitive and affective functioning. Even if the hypothetical pathways involving empathy and schizotypy were assumed to be different than those in this study, the data would still indicate important relationships with social connectedness, and accordingly, the potential value of improving social connectedness by targeting impairments associated with empathy. For practical purposes, social connectedness is an intervention target with unquestionable importance, functionally proximal to poor mental and physical health. The cognitive and affect regulation processes associated with empathy are in turn proximal to social connectedness, however complex the pathways among them, whether direct or indirect, mediated or moderated; therefore, they remain candidates as targets for intervention. As with etiological vulnerability in children, research supports targeting cognitive and affect regulation impairments in treating high risk individuals (e.g., [100]) and post-onset adults (e.g., [101]).

A secondary limitation of this study is its use of a continuous, nomothetic conceptualization of schizotypy. Whether schizotypy is a continuous dimension with meaningful variability even in normative subpopulations, or a categorical condition detected by extreme elevations on continuous psychometric measures, or both, has been controversial since the concept appeared in the early 1960s [22,102,103,104,105,106]. The significance of relationships observed across the entire range of schizotypy measures could be entirely different from those that most distinguish vulnerable individuals from the rest of the population. A more focused study of a schizotypy subpopulation, probably no more than 10% of the general population [102], brings methodological challenges that make complex multivariate analyses like those in the present study more difficult. Pending large-scale multivariate studies focused on such a small subpopulation, the present data do provide some limited opportunities to infer whether the implications of variability in the normative range of the sample are comparable to those in a more extreme range, with respect to empathy and social connectedness. This is a logical next step for research on this topic. However, until the methodological challenges are met, and it becomes known whether schizotypy is different in “healthy” vs. “clinical” samples, even when measured with the same instrument, the applicability of findings across groups is somewhat conjectural. Meanwhile, neither assumption invalidates the correlations observed in this study, nor compromises the potential significance of therapeutic interventions targeting the identified pathways in healthy but vulnerable individuals.

This study’s use of a college student sample arguably creates a similar limitation, but a minor one at most. There is no reason to assume that the current findings generalize to other populations. However, it is unclear whether the study of other samples of participants with the same demographics except for college status would advance the science, if that were even possible. As in the findings of this study, subtypes and subpopulations within schizotypy are important, and much progress lies in understanding both commonalities and differences. College students are not simply a “sample of convenience” for schizotypy research. They are of interest in their own right, due to the relevance to schizotypy of their demographics, their stage of development, their range of personal and social functioning, and other characteristics. There is value in understanding how this works in college students, as well as how it works differently in other groups. As our theories progress, we should expect them to incorporate both.

Finally, as discussed in the Introduction, it is a limitation that this study, and the preponderance of empathy research, includes no task-performance-based constructs or measures. A few laboratory measures have shown promise in subjecting the relevant cognitive and affective processes to experimental analysis, and as our understanding of empathy in the “normal” context progresses, and as the world emerges (hopefully) from pandemic conditions that inhibit laboratory procedures, inclusion of such measures will be a key development in schizotypy research.

## 6. Conclusions

The present findings illuminate key cognitive and affective processes associated with schizotypy that compromise social connectedness, a resiliency factor that sustains quality of life and moderates consequences of health vulnerabilities in people with schizophrenia spectrum disorders. The pathways among these processes and their association with schizotypy in college students provide important clues to the etiology of impaired social connectedness observed in clinical populations. The three dimensions or categories of schizotypy, positive, negative, and disorganized, represent distinct and different pathways to cognitive impairments and social disconnectedness. Treatment interventions that seek to improve the health and quality of life of people with severe mental illness by increasing their social connectedness are expected to be enhanced by components that directly address specific processes associated with empathy, including alexithymia, distress tolerance, emotion contagion, cognitive empathy, and responsiveness to others. The development of interventions to address these vulnerabilities has already begun. The multiplicity and complexity of relationships identified in this study and previous research reinforce familiar principles of treatment and rehabilitation for SMI, that an optimal outcome requires integrated, interdisciplinary assessment and intervention planning, targeting specific impairments and vulnerabilities, but with the ultimate goal of achieving a holistic recovery and maximum quality of life.

## Figures and Tables

**Figure 1 behavsci-12-00253-f001:**
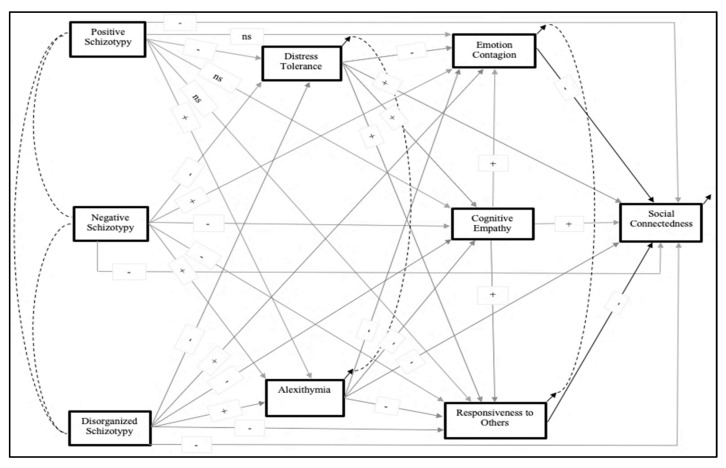
Hypothesized pathways between 3 subtypes of schizotypy and social connectedness *. * “ns” indicates the path coefficient is hypothesized to be zero (not statistically significant); “+” and “-” indicate the valence of the path coefficient, reflecting whether higher scores on the operational measure are presumed to be positive or negative influences.

**Figure 2 behavsci-12-00253-f002:**
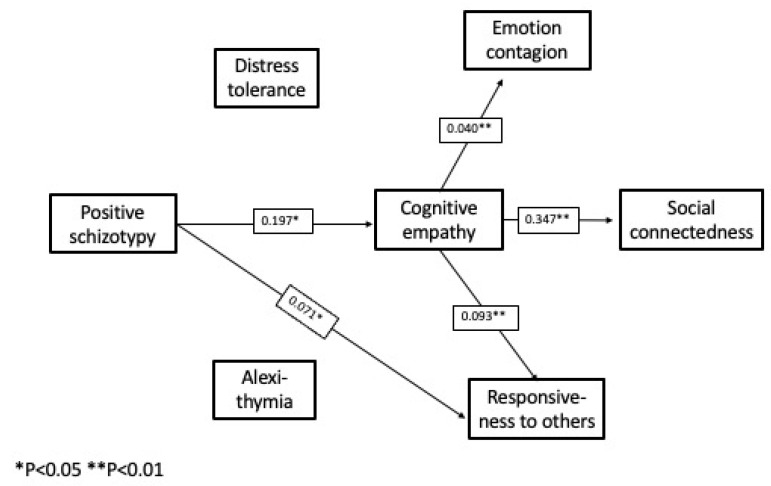
Computed pathways between positive schizotypy and social connectedness (unstandardized path coefficients).

**Figure 3 behavsci-12-00253-f003:**
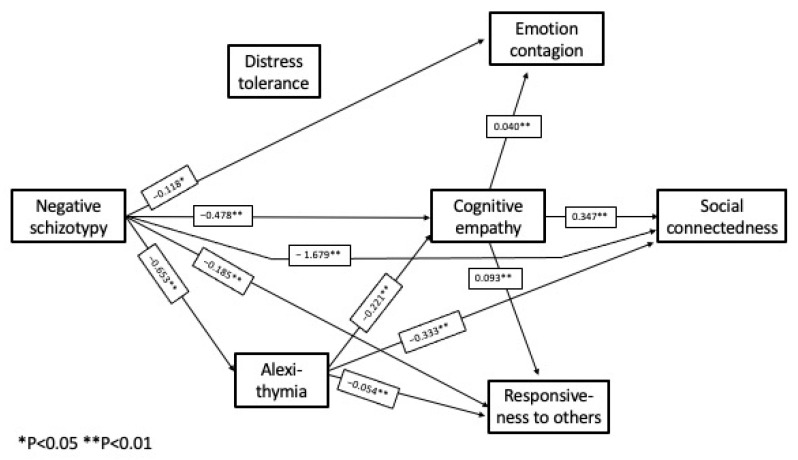
Computed pathways between negative schizotypy and social connectedness (unstandardized path coefficients).

**Figure 4 behavsci-12-00253-f004:**
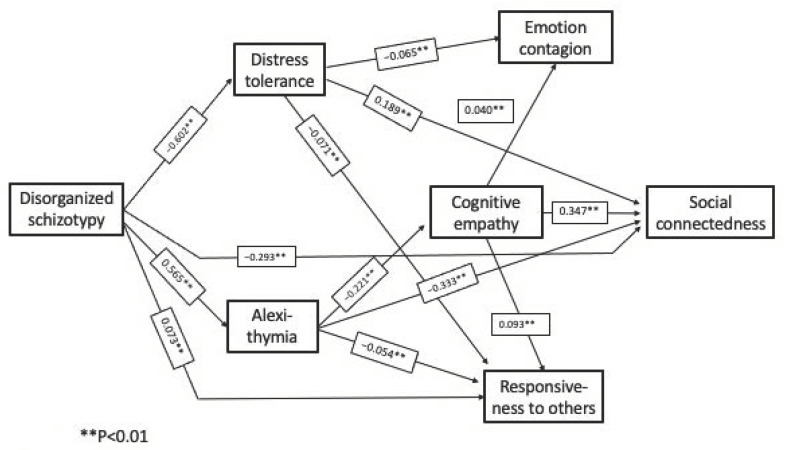
Computed pathways between disorganized schizotypy and social connectedness (unstandardized path coefficients).

**Table 1 behavsci-12-00253-t001:** Participant demographics.

	N	%
Sex at Birth		
Male	158	19.2
Female	664	80.6
Gender Identity		
Male	155	18.8
Female	654	79.4
Genderfluid	6	0.7
Transgender (Female to Male)	3	0.4
Other	6	0.7
Race		
White	641	77.8
Black	42	5.1
Asian	73	8.9
American Indian or Alaska Native	5	0.6
Pacific Islander	2	0.2
Multiracial	36	4.4
Other	22	2.7
Ethnicity		
Hispanic/Latinx/Spanish	89	10.8
Marital Status		
Married or Domestic Partnership	10	1.2
Single/Unmarried	812	98.5

**Table 2 behavsci-12-00253-t002:** Means and standard deviations of study measures.

	M	SD	N
MSS			
Positive	3.16	4.252	824
Negative	3.25	3.945	824
Disorganized	3.66	5.461	824
QCAE			
Cognitive Empathy	61.39	8.284	824
Emotion Contagion	12.26	2.280	824
Responsiveness to Others	21.068	3.468	824
DTS	48.98	12.297	823
TAS-20	47.23	12.046	822
SCS-R	87.04	16.908	822
MCSD-SF-C	6.49	2.747	822
CECQ	8.57	4.874	822

Note. MSS = The Multidimensional Schizotypy Scale; QCAE = The Questionnaire of Cognitive and Affective Empathy; DTS = The Distress Tolerance Scale; TAS-20 = The 20 Item Toronto Alexithymia Scale; SCS-R = The Social Connectedness Scale-Revised; MCSD-SF-C = The Marlowe–Crowne Social Desirability Scale-Short Form Version C; CECQ = The Controlling for Environmental Confounds Questionnaire.

**Table 3 behavsci-12-00253-t003:** Correlation matrix of the study measures.

	*1*	*2*	*3*	*4*	*5*	*6*	*7*	*8*
1. MSS-Positive								
2. MSS-Negative	0.431 *							
3. MSS-Disorganized	0.610 *	0.490 *						
4. QCAE-Cognitive Empathy	−0.112 *	−0.322 *	−0.184 *					
5. QCAE-Emotion Contagion	0.114 *	−0.140 *	0.132 *	0.162 *				
6. QCAE-Responsiveness to Others	0.046	−0.232 *	0.038	0.308 *	0.418 *			
7. DTS	−0.267 *	−0.155 *	−0.353 *	0.152 *	−0.384 *	−0.205 *		
8. TAS-20	0.372 *	0.404 *	0.455 *	−0.381 *	0.060	−0.154 *	−0.436 *	
9. SCS-R	−0.389 *	−0.614 *	−0.501 *	0.419 *	−0.046	0.106 *	0.408 *	−0.588 *

Note. * *p* < 0.001 (two-tailed), surviving Bonferonni correction for *p* < 0.05. MSS = The Multidimensional Schizotypy Scale; QCAE = The Questionnaire of Cognitive and Affective Empathy; DTS = The Distress Tolerance Scale; TAS-20 = The 20 Item Toronto Alexithymia Scale; SCS-R = The Social Connectedness Scale-Revised.

**Table 4 behavsci-12-00253-t004:** Significant indirect effects in the complete path model (unstandardized path coefficients).

Independent Variable	Dependent Variable	Mediating Variable(s)	Path Coefficient	95% CI *
Positive schizotypy	Emotion contagion	Cognitive empathy	0.008	0.002, 0.018
Positive schizotypy	Responsiveness to others	Cognitive empathy	0.018	0.004, 0.037
Positive schizotypy	Social connectedness	Cognitive empathy	0.068	0.015, 0.136
Negative schizotypy	Cognitive empathy	Alexithymia	−0.144	−0.212, −0.091
Negative schizotypy	Emotion contagion	Cognitive empathy	−0.019	−0.036, −0.008
Negative schizotypy	Emotion contagion	Alexithymia, Cognitive empathy	−0.006	−0.011, −0.003
Negative schizotypy	Responsiveness to others	Alexithymia	−0.035	−0.059, −0.018
Negative schizotypy	Responsiveness to others	Cognitive empathy	−0.044	−0.070, −0.025
Negative schizotypy	Responsiveness to others	Alexithymia, Cognitive empathy	−0.013	−0.022, −0.008
Negative schizotypy	Social connectedness	Alexithymia	−0.217	−0.338, −0.128
Negative schizotypy	Social connectedness	Cognitive empathy	−0.166	−0.259, −0.099
Negative schizotypy	Social connectedness	Alexithymia, Cognitive empathy	−0.050	−0.084, −0.029
Disorganized Schizotypy	Cognitive empathy	Alexithymia	−0.125	−0.177, −0.085
Disorganized Schizotypy	Emotion contagion	Distress tolerance	0.039	0.026, 0.055
Disorganized Schizotypy	Emotion contagion	Alexithymia, Cognitive empathy	−0.005	−0.009, −0.002
Disorganized Schizotypy	Responsiveness to others	Distress tolerance	0.043	0.027, 0.063
Disorganized Schizotypy	Responsiveness to others	Alexithymia	−0.030	−0.049, −0.016
Disorganized Schizotypy	Responsiveness to others	Alexithymia, Cognitive empathy	−0.012	−0.018, −0.007
Disorganized Schizotypy	Social connectedness	Alexithymia	−0.188	−0.271, −0.121
Disorganized Schizotypy	Social connectedness	Distress tolerance	−0.114	−0.184, −0.063
Disorganized Schizotypy	Social connectedness	Alexithymia, Cognitive empathy	−0.043	−0.070, −0.026

* Confidence Interval for *p* < 0.05 (two-tailed).

## Data Availability

Not applicable.

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
