# Peer review of "Social Connectedness in Schizotypy: The Role of Cognitive and Affective Empathy"

_behavsci, 2022, doi:10.3390/bs12080253_

Round 1
Reviewer 1 Report
The presented article deals with a current and interesting topic. This topic offers a look at schizophrenia in relation to social issues. In the article, I appreciate the relatively good insight into the issue. The introduction of the article is very well done using relevant citation sources.
However, it is relatively unclear from the submitted document where the introduction ends. I miss the logical division eg Introduction and Background. The text would deserve a deeper division into these categories. After all, even the authors direct the text from the general to the specific. So there is a need to clearly separate and define these areas. So I recommend to divide the introduction and clearly show what is really a general introduction and what is already a specific background of the article. This will be followed by a description of the research set, the research problem and the goals that I have probably overlooked or are not clearly formulated in the work. The chapter results is then professionally processed from my point of view. The results are interpreted appropriately. Statistical procedures were appropriately chosen together with the presentation of Chronbach's alpha. What statistical program was used? I recommend adding, but this is really a slight shortcoming with regard to the very quality of the study's outputs. I also consider the corelation of the study to be professional. I dare say that I do not find any significant shortcomings here that should prevent the publication of the article.
The discussion is again a quality work of the authors, but I lack more support for citation sources. After all, the discussion is a tool for comparing, refuting, incorporating the results of the work into the already current results of other authors, etc. Here are some sources, but I personally would use the discussion to cover and present the results. However, I will leave it to the authors to decide whether or not they want to enrich the discussion. Personally, I have no problem with this version of the discussion and I am willing to confirm it for publication in this form.
However, what I miss and request for addition is a conclusion. I did not find this final chapter in the presented version of the article. I recommend focusing on general topics in the field of psychiatric rehabilitation and pointing out the possibilities of, for example, multidisciplinary teams and thus concluding the survey, which the authors carried out. For example, the article https://www.webofscience.com/wos/woscc/full-record/WOS:000545292800010 seems to be appropriate. In conclusion, I recommend focusing on the needs of comprehensive approaches to solving the problem. Therefore, please add at least one paragraph to the conclusion. The article can be taken as a guide, but after searching for keywords on the topic, this one is published.
After completing the conclusion, I am inclined to publish this article, as I find it really high quality and necessary. The article has a very good impact and is a benefit for both the scientific and lay community.
Author Response
Dear Reviewer,
Thank you for your thoughtful comments and feedback. In response, we have made a number of changes beginning in the introduction. This section has been shortened and reorganized, such that we have tried to make a more logical distinction between general introduction and relevant background. We have also attempted to end this section with a more detailed description of gaps in current literature that set up our study aims and hypotheses. We have also taken your feedback in the methods section and added the statistical program used to calculate Cronbach’s alpha.
We appreciate the comment to add a conclusion section to our manuscript. This is now included in our edits with some additional thoughts on how our study supports ongoing interdisciplinary assessment and treatment planning that will aid in the recovery of persons with SMI.
Kind regards,
Jessica Stinson
Reviewer 2 Report
11. Abstract should include sample size and a brief description of schizotypy, including the three schizotypy dimensions.
22. The structure of the manuscript could be improved. In the introduction, study aims should come towards the end, followed by the hypotheses. Both the introduction and discussion would benefit from sub-headings.
33. The Introduction would benefit from a major revision. It is too long and needs to be condensed. It also reads at times like a list of definitions and studies – the information needs to be better integrated with each other. There needs to be clear justification as to why each construct has been chosen for this study. You should briefly discuss the previous research in schizotypy/schizophrenia-spectrum disorders, and then identify gaps to highlight the novelty of your study.
44. The Bonferroni corrections and comments about normality should be mentioned in the data analysis section.
55. You need to make it clear in the demographics table that some are frequencies and percentages. Also, tables should go after the paragraph it is mentioned in, not before.
66. Were students with mental health disorders included in the study? A snapshot of mental health state would aid in the interpretation of findings.
77. Please include the means and SDs for the different variables. For example, it’s important to know if this was a high or low schizotypy population.
88. The findings should be more cautiously interpreted. There could be a small section about how these findings relate to schizophrenia, but this needs to be toned down as this is a healthy sample.
Author Response
Dear Reviewer,
We thank you for your thoughtful contributions to the editing of our manuscript. Please see below for our response to your comments:
Item 1: We have added the sample size and a brief description of the dimensions of schizotypy to our abstract.
Item 2: Study aims and hypotheses are now located at the end of the introduction section and subheadings have been added to both the introduction and discussion sections.
Item 3: The introduction section has been shortened and reorganized in a manner in which we feel better integrates this information. We have also enriched the introduction with additional discussion of the gaps in current literature that inform the current study and clarify our aims.
Item 4: Bonferroni corrections and normality tests are now mentioned in the data analysis section.
Item 5: The demographics table has been corrected to now reflect frequencies and percentages. Tables and figures have been moved to after the paragraphs in which they are discussed.
Item 6: The following was added to the description of participants: “No screening was performed for elevated schizotypy or mental health history. The distributions of these indicators in this sample are presumed to be the same as for university undergraduates in general.”
Item 7: A new table was added reflecting the means and standard deviations of each of the primary study variables.
Item 8: We understand this concern to be part of the general limitation of using a nomothetic measure of schizotypy. Whether the construct is “really” nomothetic or categorical, or whether it is different in “healthy” vs. “unhealthy” samples, even when measured by the same instrument, cannot be determined in this study. This is a logical next step in the research process. One could arguably address the question by asking whether the parameters and correlations in the 90th %tile of this sample are the same as in the rest of the sample, but almost by definition the former subsample is too small to provide sufficient power to answer the question. Similarly, there is no unquestionable rationale for assuming that the 90%tile of a schizotypal distribution in a “healthy” sample is comparable to individuals with the same scores in a “clinical” sample, even though that is the common presumption among schizotypy researchers, and is arguably a logical requirement for the validity of the construct. If schizotypy is truly a vulnerability or risk factor for frank schizophrenia, we must expect it to be present in “healthy” populations at a rate even higher than the rate of frank illness (this was Meehl’s key assumption). Either way, neither assumption undermines the correlations observed in this study, or compromises the potential significance of therapeutic intervention. We have expanded our discussion of how the nomothetic assumption limits interpretation of the present findings in this respect. The concern is further discussed in the following paragraph about the limitations of a “healthy” student sample.
Kind regards,
Jessica Stinson
Round 2
Reviewer 2 Report
The authors have addressed the comments well. No further changes requested.